# Preferences for accessing sexual health services among middle-aged and older adults in the UK: a study protocol for a discrete choice experiment using mixed methods

Eneyi Kpokiri ![ORCID] ,[1] Stephen W. Pan,[2] Jason J. Ong ![ORCID] ,[1,3] Emily Greaves,[1] Junead Khan,[1] Sophie Bowen,[1] Tracey Jannaway,[4] Fern Terris-Prestholt,[5] Clare Tanton ![ORCID] ,[6] Hannah Kuper ![ORCID] ,[5] Thomas Shakespeare,[5] Joseph D. Tucker,[1,7] Dan Wu ![ORCID] [1]

EK and SWP contributed equally.

For numbered affiliations see end of article.

**Correspondence to**
Dr Dan Wu;
dan.wu@lshtm.ac.uk

## ABSTRACT

**Introduction** Sexual health is essential for general health and well-being. Sexual health services for middle-aged and older adults are not prioritised and optimising available services for this population is often overlooked. Not much is known about preferences for accessing sexual health services among middle-aged and older people or level of satisfaction with current services. The aim of this study is to explore preferences for seeking sexual health services among middle-aged and older adults in the UK. This study will use discrete choice experiments (DCEs) including initial qualitative interviews followed by the survey, which have been used as a tool to explore preferences in various health service delivery.

**Methods and analysis** The project will be carried out in two phases. First, we will conduct in-depth semi-structured interviews with 20–30 adults (aged 45+), including disabled people, and those from sexual minority groups resident in the UK. Interviews will explore indications, preferences and factors related to accessing sexual health services. Themes and subthemes emerging from the analysis of the interviews will then be used to design the choice sets and attribute level for the DCEs. For the second phase, for the DCEs, we will design choice sets composed of sexual health service delivery scenarios. The software Ngene will be used to develop the experimental design matrix for the DCE. We will use descriptive statistics to summarise the key sociodemographic characteristics of the study population. Multinomial logit, latent class and mixed logit models will be explored to assess sexual health service preferences and preference heterogeneity.

**Ethics and dissemination** Ethical approval for both parts of this study was granted by the Research and Ethics Committee at the London School of Hygiene & Tropical Medicine. Findings from this study will be disseminated widely to relevant stakeholders via scheduled meetings, webinars, presentations and journal publications.

## STRENGTHS AND LIMITATIONS OF THIS STUDY

⇒ This is the first discrete choice experiment (DCE) exploring preferences for sexual health services among middle-aged and older adults including disabled people.
⇒ Our DCE is designed based on detailed formative research whereby choice sets are developed from emerging themes from 21 semi-structured interviews.
⇒ While the survey targets mature adults (age 45 and older), we sought to ensure inclusion of those from sexual minority groups and disabled people by recruiting from local community-based organisations. This is to ensure diversity within this population to allow for subanalysis.
⇒ Recruitment will be restricted to only adults aged 45 and older and living in the UK as a result, generalising the findings may be restricted to only the UK or other places with similar settings.

## INTRODUCTION

Sexual health is an integral part of health, well-being and quality of life. However, many existing sexual health programmes, interventions and research are tailored for youth and younger adults.[1] The sexual health needs of middle-aged and older adults are often neglected, including by health professionals.[2] There are also a host of diverse challenges and barriers that interplay in how and why this age group access sexual health services. Older adults are increasingly experiencing higher rates of comorbid illnesses further complicating their sexual health and well-being. Up to 9% increase in sexually transmitted infections among adults aged 50 and above have been reported. The rate of HIV infections is increasing faster among people aged 50 and above than in the under 40s, consequently, the number of people aged 50 and over living with HIV has now doubled in the last decade.[3 4] Older adults

typically turn to family and friends to seek support for health needs and a study assessing sexual health clinic attendance in Britain found that non-attendance was higher among older adults.[5 6] Accordingly, there is suboptimal uptake of sexual health services among middle aged and older adults.[7 8] Within the broader group of 'older people', certain groups may be particularly neglected, such as disabled older people, as they face additional barriers to accessing health services.[9] This is an important issue, as almost one in five people in the UK are aged 65+, and 42% of people in this age group report a disability.[10] Disability is defined under the Equality act of 2010 as having a physical or mental impairment that has substantial and long-term negative effect on the ability to do normal daily activities.[11] Despite the growing population of middle-aged and older adults and an increasing body of evidence suggesting sexual health is important to quality of life, there is limited evidence to support preferences and tailor services for specific groups within this population.[12 13]

This study will explore preferences for seeking sexual health services among middle-aged and older adults. This is part of a larger study to inform areas for improved sexual health services for adults aged 45 and above, including disabled ones and from sexual minority groups living in the UK. We have chosen 45 years as our cut-off age as menopause, erectile dysfunction and other sexual health issues become more common in that age group. Our study findings will explore issues related to service uptake and identify strategies to increase the uptake of sexual health services among adults aged 45 and above in the UK.

Discrete choice experiments (DCEs) are a quantitative approach to measure strength of preferences. It is an attribute-based measure of utility, based on the assumptions that healthcare interventions, services or policies, can be described by their attributes and that user preferences depend on the levels of these attributes.[14] DCEs ask participants to make choices from hypothetical alternatives. They are increasingly used in health research to study patient and/or physician preferences.[15] For instance, DCEs have been used to determine preferences for HIV prevention and testing among a range of specific populations and settings to inform strategies for optimising health services.[16–21] DCEs have also been employed to study preferences for social care among older adults.[22] However, to date, no DCE studies have been conducted to determine preferences for sexual health services among older adults. Our study will use DCEs to explore preferences for accessing sexual health services and any heterogeneity that exists among middle-aged and older adults in the UK.

## Overall aim and objectives of this study

The main aim of this study is to identify strategies to improve access of sexual health services among adults aged 45 and above including sexual minorities and disabled people, in the UK. Specific objectives include:

1. To identify barriers and facilitators for accessing sexual health services among adults aged 45 and above including sexual minorities and disabled people, in the UK using semi-structured interviews
2. To explore key preferences for accessing services for sexual health among middle-aged and older adults in the UK using a DCE survey.

## METHODS AND ANALYSIS
### Overview of approach and methods

The research will be conducted in two phases. First, we will organise semi-structured interviews with adults aged 45 or above to better understand their preferences and the importance of sexual health service and message sharing attributes in influencing their use of these services. Interview data will be analysed thematically, and emerging themes will inform which attributes and levels will be included in the choice sets for the DCEs.

### Patient and public involvement

A key part of this study is community engagement as all studying findings will be dependent on response and input from the community members. With the aim of improving access to services, resulting strategies and solutions will be well received and more sustainable with sound community input. Consequently, community representatives have been recruited and engaged in the study design through local community-based organisations and reaching out through general practitioner (GP) service. Community-based organisations contacted where those specifically providing services to middle aged and older adults, disabled people and those from sexual minority groups. These will be continuously engaged in our survey distribution and participant recruitment, data interpretation, reporting or dissemination plans of our research.

### Phase I—semi-structured interviews

DCEs are increasingly being recommended in health research to explore patient and public preferences.[22] This includes undertaking a conjoint analysis where related characteristics will be identified early on using literature reviews, group discussions or individual interviews.[23] First, we will conduct between 20 and 30 semi-structured individual interviews with adults aged 45 and above and stop when data saturation is reached. This formative qualitative research will identify factors that are considered in the decision-making processes among this population related to sexual health services (eg, awareness, location, access process, quality of care or qualifications of healthcare providers, cost), identifying barriers and facilitators to service usage and sexual health message sharing based on older adults' perceptions of and/or experiences with sexual health services and interaction with sexual health messages. We will also field test the questionnaire using cognitive interviews prior to official start to help us understand how older individuals mentally process and respond to survey questions and DCE choice sets.

Participants will be purposely selected across different locations, age range, ethnic backgrounds and disability status. Recruitment will be via community-based organisations. We will create and disseminate a simple survey to collect data on profile including age, gender, ethnic background location in the UK, sexual orientation, relationship and disability status. We will reach out to community-based organisations to disseminate this survey within their members and networks. From the responses received we will then contact participants purposively selected to ensure good spread and diversity in the sample recruited for interviews. Interviews will be conducted via telephone calls or teleconference using Zoom, Skype or Microsoft Teams and facilitated by an experienced qualitative researcher. Teleconference calls may be either audio or video according to participant preferences. Interviews will be conducted using a topic guide (see online supplemental appendix 1) and we will aim for approximately 30 min per interview session. Adjustments will be made as needed to promote the inclusion of disabled persons (eg, use of sign language interpretation, simple language, inclusion of breaks). Each participant will receive an honorarium worth £30 in appreciation for their participation. A consent form will be provided to participants via email, which will be signed and returned before the interview commences. Verbal consent will also be obtained right at the start of the interview. Interviews will be audio-recorded following participant consent and transcribed verbatim prior to analysis.

## Qualitative data analysis

We will undertake thematic analysis of the qualitative interview data using NVivo. A codebook will be developed both inductively and deductively to guide analysis. We will employ a framework approach and transcribe all interviews verbatim then commence coding of each transcript. Codes will be aggregated to build up the emerging themes. Outcomes of qualitative data analysis will be the indications and use of sexual health services, preferences and factors affecting their decision-making to seek intimate and/or sexual relationships, barriers and facilitators to sexual health service usage. This qualitative data generated will be used to inform the DCE survey to be disseminated to participants in phase II. We will also disseminate the data as presentations in scientific conferences and publications in open access journals (see online supplemental appendix II for some preliminary results).

## Data management

All data (qualitative and quantitative) collected during the course of the research will be stored and managed securely online using our institute's Open Data Kit (ODK) at the London School of Hygiene & Tropical Medicine (LSHTM). An ODK server hosted at LSHTM allows researchers to submit de-identified data electronically from anywhere in the world to a safe and secure server environment that is hosted on our institutional information technology infrastructure. The identity of participants will be known by the researchers who will assign participant numbers to anonymise responses. Responses will be de-identified prior to analysis, and results will be reported at an aggregate level, individual quotes when used will be followed by participant assigned numbers so that participants cannot be identified.

## Phase II—discrete choice experiment survey

Second, we will use the qualitative information from Phase I to inform what attributes (characteristics) and levels (choices) will be included in the unlabeled choice set, which will be administered in an online survey to a sample of older adults to identify their preferences for accessing sexual health services. We will design choice sets composed of hypothetical sexual health service delivery scenarios. Similarly, as stated for the interview, we will employ the same strategy to recruit disabled people and people from sexual minority groups for the survey to allow subanalyses of the results. We will aim to have disabled people and sexual minorities make up 10% each out of the total sample size.

## Developing the DCE design
### Pictorial representation of attribute levels and DCE choice set size

A ranking exercise will be organised with some of the interview participants to align and finalise the attributes and levels that will be included in the final choice sets (online supplemental appendix III). After attributes and levels have been finalised, we will generate visual graphics to represent each attribute level in the DCE survey (see online supplemental appendix IV for a sample). To ensure that the visual graphics effectively represent the intended meaning of each attribute level (ie, construct validity), we will conduct qualitative think-aloud, pilot exercises with up to 10 individuals from the target population, including disabled people and those from sexual minority groups and participants will provide feedback about how they interpret and feel about the graphics.[24] This will be an iterative process with the draft pictorial concepts being tested until they are appropriately representing the intended concept. DCE surveys tend to present between 4 and 18 choice sets. During these interviews, we will also explore the optimal number of choice sets to present to each respondent to manage potential cognitive fatigue. Similar to the phase 1 interviews, participants will be purposely selected across different locations, age range, ethnic backgrounds and disability status. Again, reasonable accommodation will be made to promote inclusion of disabled persons, although people with severe cognitive/intellectual impairment will not be included in this step. Recruitment will be through collaborations with community-based organisations with which our research team already have existing research collaborations with.

## Experimental design

After visual graphics of the attribute levels have been finalised, we will use the software Ngene to develop the

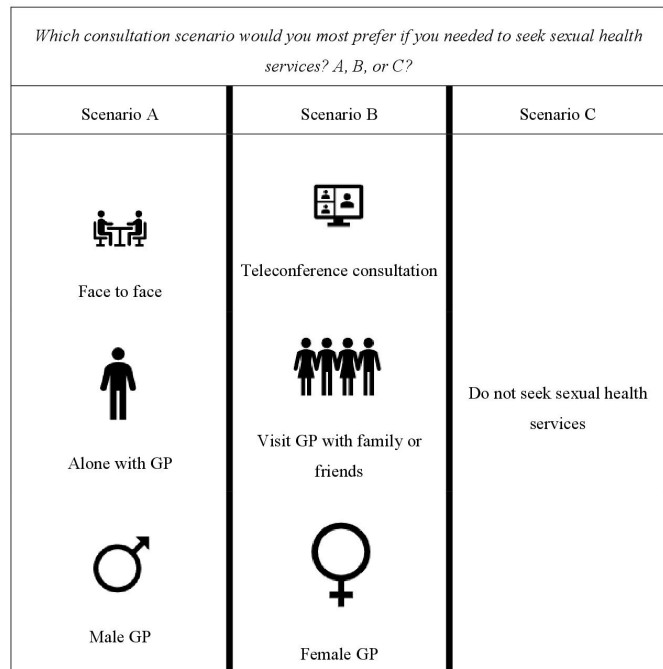

*Which consultation scenario would you most prefer if you needed to seek sexual health services? A, B, or C?*

| Scenario A | Scenario B | Scenario C |
|---|---|---|
| Face to face | Teleconference consultation | |
| Alone with GP | Visit GP with family or friends | Do not seek sexual health services |
| Male GP | Female GP | |

**Figure 1** Fictitious example of a choice set. GP, general practitioner

experimental design matrix for the DCE pilot.[25] For the pilot test, no assumptions about the parameter estimates will be made. We will then specify the appropriate number of attributes, levels, alternatives, blocks and choice sets, and Ngene will iteratively generate design matrices of increasing statistical efficiency. We will use the design with the lowest D-error generated by Ngene. An example of a choice set and DCE design is presented in figure 1. Graphics will have alt-text enabled so that they can be accessed by people with visual impairments.

### Blocking and randomisation of choice set ordering
Depending on the total number of choice sets in the DCE design, choice sets may be grouped into blocks. Respondents will then be randomly assigned to one of the blocks and asked to complete all choice sets in their block. We anticipate using between 4 and 10 blocks, with each block containing 4–10 choice sets. To guard against potential ordering effects, the order of the choice sets within each block will be randomised for each participant.

### Pilot testing
We intend to use convenience sampling to recruit up to 30 participants for the pilot test. Participants in the pilot test will receive a £20 gift card for completing the survey. Conducting the pilot test will serve two purposes: (1) identify potential problems with the DCE and survey in general (eg, cognitive fatigue or incoherent questions or lack of accessibility) and (2) generate priors (ie, parameter estimates) that will be used to develop the design of the final full-scale DCE. We will analyse the pilot data with multinomial logit (MNL) models, given the limited sample size. Priors from the pilot test will be input into

Ngene to finalise and enhance the efficiency of the full-scale DCE design.

### Full-scale DCE survey
Community-based organisations, local clinics, GP practices and older people care homes will be contacted to help distribute the online survey. Recruitment will be carried out purposively from urban and suburban areas, evenly spread across varying gender mix and the middle age and older adults age groups. We will create formats for completing the DCE surveys both online and offline.

### Eligibility criteria
We will recruit persons who meet all of the following criteria:
1. Adult aged 45 years or older.
2. Had the experience of accessing sexual reproductive health service', prior to this study.
3. Resident in the UK for at least 6 months.
4. Willing to consent to participation.

With consent, participants of the DCE will be entered into a draw where up to 20 participants will be randomly selected to receive a £100 gift card.

### Quality assessment
It is possible that some **participants** known as 'speeders' may try to complete the DCE survey as soon as possible without seriously considering the survey questions. 'Speeders' will be identified as those who complete the survey at least 50% faster than the median time for survey completion among all valid survey respondents. We will then conduct a sensitivity analysis and determine whether or not the removal of identified speeders significantly alters the results.

We will also assess the possibility of order effect bias by introducing a dummy variable for the order of the choice set. We will test for left and right bias by including a constant for the left choice and examining the extent to which the left position of the alternatives in the choice sets influences the likelihood of being chosen. Comparable probabilities of the left and centre alternatives being selected would suggest the absence of bias.

### Sample size calculation
Given the absence of any DCE priors for sexual health preferences of older adults in the UK, we used the following parametric-based DCE sample size calculation formula:

$$n \geq \frac{1-p}{rpa^2} \times \left[ (\Phi)^{-1} \left( 1 - \frac{\alpha}{2} \right) \right]^2$$

Equation 1: Parametric DCE sample size formula.[26]

Whereby *n* is the minimum sample size needed to detect *p*, the assumed true population probability of selecting uptake of sexual health services over 'opt-out', and whereby *r* is the number of choice sets that each individual completes, *a* is the level of precision around the assumed population probability, $\Phi^{-1}$ is the inverse cumulative normal distribution function and α is the

significance level.[26] Conservatively assuming that the true population probability of sexual health service uptake is 0.5 and assuming that each participant will complete eight choice sets, we estimate that a sample of 191 participants will be needed to generate a point estimate ±0.05 of the true population probabilities and with 95% confidence. However, to maintain power for two-group subgroup analyses, we will aim to recruit a total 240 participants including those from sexual minorities groups and disabled people.

## Statistical analysis

We will use descriptive statistics to summarise the key sociodemographic characteristics of the study population. We will assess sexual health service preferences and preference heterogeneity using MNL, latent class and mixed logit (MXL) models. While the MNL is the starting point for most discrete choice models, it suffers from some unrealistic assumptions, particularly around unobserved heterogeneity. The MXL model can accommodate unobserved heterogeneity. The MXL analysis will be conducted with 1000 Halton draws and will assume normally distributed preference heterogeneity for each attribute level. We will assess for unobserved heterogeneity by examining the statistical significance of the SD of the random parameters in the MXL model. We will calculate the relative importance and predicted probabilities of opt-out for select sexual health service scenarios. We will assess how sexual health service and message sharing preferences could vary among those willing and unwilling to use sexual health services by using interaction terms for sociodemographic characteristics.

We will perform latent class analyses to elucidate if there are unobserved groups of people with similar preferences for use of sexual health service.[27] First, we will conduct a latent class conditional logit model that includes individual characteristics and attribute level preferences.[27] Selection of individual characteristics will be determined by expert opinion and findings from the formative qualitative interviews. Second, we will determine the optimal number of subgroups by comparing the Bayesian information criterion (BIC) values of 2, 3, 4, 5, 6 and 7-class models and each models' interpretability and class size distribution.[27] Lower BIC values indicate better model fit. Third, we will use descriptive statistics to characterise each class based on their members' preferences and characteristics. All analyses will use effects coding and be weighted to reflect the population of middle-aged and older adults in the UK.

## Attribute relative importance and probability of opt-out

For each of the sex-class combinations, the relative importance of each attribute will be calculated by dividing the range of the parameter estimates of a given attribute by the summation of the ranges of all attributes.[28]

We will further illustrate potential preference heterogeneities by using equation 2 to calculate the predicted probabilities of selecting 'opt-out' over select sexual health service scenarios for each sex-class combination. Specifically, the predicted probability of a participant choosing alternative $i$ over alternative $j$ can be estimated by dividing the exponentiated summation of coefficients for scenario $i$ by the exponentiated summation of coefficients for scenario $j$.

$$\mathrm{Pi} = \frac{e^{\beta x_i}}{\sum e^{\beta x_j}}$$

Equation 2: Predicted probability of selecting alternative $i$ within a two-alternative choice set'.

## Impact

This study will improve our understanding of older people's preferences and factors important for sexual health services. Our data will be useful to develop sexual health services and programmes tailored for older people. By engaging older adults, this study will raise awareness for the need to access sexual health services irrespective of age and also ultimately identify strategies co-created by the end-users to increase service uptake.

## Feasibility

Our project is feasible as the team has expertise from multiple disciplines to examine sexual health among older adults with unique strengths in DCEs, disability studies and sexual health research.

## Ethics and dissemination

Ethical approval was obtained from the Research and Ethics Committee at the London School of Hygiene & Tropical Medicine, UK (26134). Findings from this study will be disseminated widely to relevant stakeholders via scheduled meetings, webinars, presentations at conferences and in peer-reviewed journal manuscripts.

**Author affiliations**
[1]Department of Clinical Research, London School of Hygiene & Tropical Medicine, London, UK
[2]Department of Health and Environmental Sciences, Xi'an Jiaotong-Liverpool University, Suzhou, Jiangsu, China
[3]Melbourne Sexual Health Centre, Monash University, Melbourne, Victoria, Australia
[4]Independent Living Alternatives, London, UK
[5]Department of Population Health, London School of Hygiene & Tropical Medicine, London, UK
[6]Department of Global Health and Development, London School of Hygiene & Tropical Medicine, London, UK
[7]University of North Carolina Project China, Guangzhou, Guangdong, China

**Contributors** DW and JT conceived the idea. DW oversees the study. EK and SWP wrote the first draft of the protocol paper. EK, EG and SB conducted the interviews, developed attributes and levels and JK helped to refine the discrete choice experiment survey design. JO, SWP, FT-P and JT provided additional advice on the discrete choice experiment design and will carry out the experiment design. TJ, CT, HK, TS and JT provided expert advice on improving inclusiveness and appropriateness of the protocol. All authors reviewed, provided constructive comments on improving the design and the manuscript. All authors have seen and approved the final version of this manuscript.

**Funding** The study is supported by the Economic and Social Research Council, UK Research and Innovation (UKRI) (grant number: ES/T014547/1).

**Competing interests** None declared.

**Patient and public involvement** Patients and/or the public were involved in the design, or conduct, or reporting, or dissemination plans of this research. Refer to the Methods section for further details.

**Patient consent for publication** Not applicable.

**Provenance and peer review** Not commissioned; externally peer reviewed.

**ORCID iDs**
Eneyi Kpokiri http://orcid.org/0000-0003-1180-1439
Jason J. Ong http://orcid.org/0000-0001-5784-7403
Clare Tanton http://orcid.org/0000-0002-4612-1858
Hannah Kuper http://orcid.org/0000-0002-8952-0023
Dan Wu http://orcid.org/0000-0003-0415-5467

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
