## [Reviewer comments · BMJ Open]

ARTICLE DETAILS

TITLE (PROVISIONAL)	Preferences for accessing sexual health services among middle-aged and older adults in the United Kingdom: a study protocol for a discrete choice experiment using mixed methods
AUTHORS	Kpokiri, Eneyi; Pan, Stephen W.; Ong, Jason; Greaves, Emily; Khan, Junead; Bowen, Sophie; Jannaway, Tracey; Terris-Prestholt, Fern; Tanton, Clare; Kuper, Hannah; Shakespeare, Tom; Tucker, Joseph; Wu, Dan

VERSION 1 – REVIEW

REVIEWER	Gebremeskel, Akalewold University of Ottawa, School of International Development and Global Studies, Faculty of Social Science, University of Ottawa
REVIEW RETURNED	11-Aug-2022

GENERAL COMMENTS	Thank you for inviting me to review this protocol: "Preferences for accessing sexual health services among older adults including those with disabilities in the UK: A Discrete Choice Experiment Protocol . Please, find comments below. Page # 2 line 5: The title of the protocol need to indicate both phases of the study, as per the content of the protocol: A Discrete Choice Experiment Protocol Page # 3 line 13: replace the word access access to 'assess' or 'explore' appropriate term Page # 3 line 16-18: "The study will utilise Discrete Choice Experiments (DCEs), which have been used as a tool to explore preferences in various health service delivery." It is not clear why you are not indicating both methods, please clarify or address both phases or methods. Page # 3 line 21: Please clarify the inclusion criteria for study population. Beyond your argument on page #4 line 22-24, the background section need to be stronger and more convincing based on the review of the existing policy and practice gaps in the context of the study population and area. For example, the team need to address the following points: What are the levels of sexual reproductive health service utilization among the study population; What are the major sexual reproductive health problem in your study area, particularly for the study population? Which policy/ interventions are working well? which are not working? What are the policy/ intervention or evidence gaps? How this study will address/ contributes to the gaps? Page # 5 line 10: "Recruitment will be restricted to only persons within the UK as a result..." Please restate using the specific study population/ group, as this study is not considering all people in UK.
---

	Page #5 line 48: What do you mean by “generate new knowledge” please restate the whole statement. How you determine the study population ‘cutpoint’ in the age and what is the rationale? Page #5 line 19: Please clarify your study population inclusion criteria, as it seems not clear yet. See page Page #5 line 19 “among older adults, including sexual minorities and those with disabilities, in the UK.” And revise the whole document accordingly. You could also consider: All(general) Older adult (with specific cutpoint in age, without restriction based on their diver sexual orientation and disability status) or Older adult + disability + sexual minority/orientation (with restriction). Page 5 line 16: “Study aims and objectives”. And Please indicate the main objective of each phase of the study. The heading itself and its description is even not clear, pleas proof read and rewrite. Page # 5 line 22: “To use DCE to identify preferences related to accessing services for sexual health” Please restate the objective in terms of the research outcome, not the research method, ”DCE”. Revise the whole document accordingly. Page #5 line 44: “Patient and Public Involvement” Please provide the importance/ rationale of community engagement for this study and how community representatives will be recruited, Beyond facilitating the data collection/ recruitment, do you have any collaboration plan with them as a potential knowledge user? Page # 5 under “Phase I- Semi-structured interviews” Please provide the appropriate justification with citation for the rationale to use semi-structured interview prior to conducting survey. Page 7 line 15-18: Please provide more clarification on the procedures you will follow to purposively selected participants by their age range, ethnic backgrounds, and disability status’ as some of these information’s are personal. Are you planning to use the profile of the potential participants organized by the community organizations or others? if so how you would access them? What would be the role of community-based organisations for this study? Please indicate when and where the consent forms will be signed? Please clarify how the data will be managed (qualitative and quantitative) to ensure confidentiality. Page 7 line 25: What kind of software are you planning to use for the thematic analysis? For example, you could consider NVivo or other. What is your plan for the qualitative data dissemination? Page 8 line 27: “Similar to the phase 1 interviews, participants will be purposively selected across different...” see my comment for Page 7 line 15-18 above. And provide sufficient justification. Page #9 line 50: Seems personal /private, I think you have to re state “Have had sexual health problems, low sexual satisfaction or function in the last five years”, maybe you can use the phrase like ‘if they have the experience of accessing sexual reproductive health service’, prior to this study. Page #9 line 56: Change the word “Able” to “willing to” Page #11 line 16: Under “Impact”, please state the impact of the study for the study population or for the knowledge users. Page 13: References section require formatting.
--	---

REVIEWER	Curley, Christine University of Connecticut
REVIEW RETURNED	31-Aug-2022

GENERAL COMMENTS

The need for this study is clear, as sexual health for older adults is often not discussed and understanding the sexual health concerns of older adults is paramount to delivering appropriate, accessible, and effective services. Overall, this study protocol is well thought out and presented.

In the abstract, the two step methods are not clear; it appears that the first stage is the 20-30 qualitative interviews and the second stage is a DCE survey design. Will this be conducted with a second (larger) group of participants or the same 30 participants? Clearly state the components of Phase II in the abstract.

Regarding the sample, please indicate how many disabled participants you would be seeking and consider how disability is defined, whether it means physical disabilities, intellectual disabilities or any other disabilities, as the sexual health of these individuals needs may differ.

In addition, consider whether defining 45+ as “older adults” is appropriate as in most developed countries 45 is closer to middle age. If you have selected 45 as an age where sexual health concerns increase due to menopause, erectile dysfunction or the like, clearly state the basis for this decision.

On page 4 please “GIVE AN EXAMPLE” as this seems to be missing.

Phase II – DCE Survey (p 6) – what is the intended sample size for the described online survey of older adults? Justification for this sample size looks sufficient as discussed on page 9; however it would be clearer if sample size was listed here as well. Pilot sample is discussed on page 8 -is 30 participants; is this sample appropriate and will it be inclusive in the same way as the qualitative Phase I portion?

Under “Developing the DCE Design” I am not convinced that 3-5 participants will be sufficient to judge the visual graphics given the fact that the researchers are seeking to design protocols for a range of sexualities, demographics, and ability levels. Perhaps you can be more specific as to who the characteristics of these 3-5 participants.

On pages 9-10 statistical analysis is clearly described. Please provide similar detail for the qualitative portion of this study. I will note that Table 2 in the supplemental materials appears to be on track.

This is an important study and I applaud the researchers for undertaking this work.

VERSION 1 – AUTHOR RESPONSE

Reviewer: 1

Comments to the Author:

Thank you for inviting me to review this protocol: “Preferences for accessing sexual health services among older adults including those with disabilities in the UK: A Discrete Choice Experiment Protocol Please, find comments below.

Comment 1: Page # 2 line 5: The title of the protocol need to indicate both phases of the study, as per the content of the protocol: A Discrete Choice Experiment Protocol

Response: We revised the title to “Preferences for accessing sexual health services among middle-aged and older adults in the UK: a study protocol for a discrete choice experiment using mixed methods”

Comment 2: Page # 3 line 13: replace the word access access to ‘assess’ or ‘explore’ appropriate term

Response: Thank you for spotting the error. The word access has now been replaced with explore.

Comment 3: Page # 3 line 16-18: “The study will utilise Discrete Choice Experiments (DCEs), which have been used as a tool to explore preferences in various health service delivery.” It is not clear why you are not indicating both methods, please clarify or address both phases or methods.

Response: We have now clarified that the DCE includes initial qualitative interviews followed by the survey administration.

Comment 4: Page # 3 line 21: Please clarify the inclusion criteria for study population.

Response: We have clarified the inclusion criteria to be adults aged 45 and above resident in the UK and include disabled people and those from sexual minority groups.

Comment 5: Beyond your argument on page #4 line 22-24, the background section need to be stronger and more convincing based on the review of the existing policy and practice gaps in the context of the study population and area. For example, the team need to address the following points: What are the levels of sexual reproductive health service utilization among the study population; What are the major sexual reproductive health problem in your study area, particularly for the study population? Which policy/ interventions are working well? which are not working? What are the policy/ intervention or evidence gaps? How this study will address/ contributes to the gaps?

Response: Thank you for providing these useful insights to further strengthen the background section. We have now added in some rationale for older adults requiring sexual health services, highlighting major SRH issues for this population and providing some evidence pointing to lower service uptake among the group and in the study location. The related references have been added in also.

Comment 6: Page # 5 line 10: “Recruitment will be restricted to only persons within the UK as a result...” Please restate using the specific study population/ group, as this study is not considering all people in UK.

Response: Thank you for identifying this omission. The sentence has been revised to “recruitment will be restricted to only adults aged 45 and older and living in the UK”

Comment 7: Page #5 line 48: What do you mean by “generate new knowledge” please restate the whole statement. How you determine the study population ‘cutpoint’ in the age and what is the rationale?

Response: Thank you for pointing this out. We have chosen 45 years as our cut off age as menopause, erectile dysfunction and other sexual health issues become more common in that age

group. The has now been revised to “We have chosen 45 years as our cut off age as menopause, erectile dysfunction and other sexual health issues become more common in that age group. Our study findings will explore issues related to service uptake and identify strategies to increase the uptake of sexual health services among adults aged 45 and above in the UK”.

Comment 8: Page #5 line 19: Please clarify your study population inclusion criteria, as it seems not clear yet. See page Page #5 line 19 “among older adults, including sexual minorities and those with disabilities, in the UK.” And revise the whole document accordingly. You could also consider: All(general) Older adult (with specific cutpoint in age, without restriction based on their diver sexual orientation and disability status) or Older adult + disability + sexual minority/orientation (with restriction).

Response: This has now been clearly stated as adults aged 45 and above, including disabled people and from sexual minority groups living in the United Kingdom, in this instance and across the entire document.

Comment 9: Page 5 line 16: “Study aims and objectives”. And Please indicate the main objective of each phase of the study. The heading itself and its description is even not clear, pleas proof read and rewrite.

Response: Thanks for identifying some inconsistencies here. The entire section has now been revised as below:

Overall aim and objectives of this study

The main aim of this study is to identify strategies to improve access of sexual health services among adults aged 45 and above including sexual minorities and disabled people, in the UK. Specific objectives include:

- 1) To identify some barriers and facilitators for accessing sexual health services among adults aged 45 and above including sexual minorities and disabled people, in the UK using semi-structured interviews
- 2) To explore key preferences for accessing services for sexual health using a DCE survey

Comment 10: Page # 5 line 22: “To use DCE to identify preferences related to accessing services for sexual health” Please restate the objective in terms of the research outcome, not the research method, “DCE”. Revise the whole document accordingly.

Response: This has been revised as captured in the response above.

Comment 11: Page #5 line 44: “Patient and Public Involvement” Please provide the importance/ rationale of community engagement for this study and how community representatives will be recruited, Beyond facilitating the data collection/ recruitment, do you have any collaboration plan with them as a potential knowledge user?

Response: We have now revised this section including the rationale for community engagement and how we exactly recruited the community engagement group.

Comment 12: Page # 5 under “Phase I- Semi-structured interviews” Please provide the appropriate justification with citation for the rationale to use semi-structured interview prior to conducting survey.

Response: We have clarified the stages of DCE involving a conjoint analysis where related characteristics will be identified early on using literature reviews, group discussions or individual interviews, Appropriate references have been cited

Comment 13: Page 7 line 15-18: Please provide more clarification on the procedures you will follow to purposively selected participants by their age range, ethnic backgrounds, and disability status’ as some of these information’s are personal. Are you planning to use the profile of the potential participants organized by the community organizations or others? if so how you would access them? What would be the role of community-based organisations for this study?

Response: We have now clarified the role of the CBOs and how we will ensure diversity in the participants recruited. This section has been updated with this text: We will create and disseminate a simple survey to collect data on profile including age, gender, ethnic background location in the UK, sexual orientation, relationship, and disability status. We will reach out to community-based organisations to disseminate this survey within their members and networks. From the responses received we will then contact participants purposively selected to ensure good spread and diversity in the sample recruited for interviews.

Comment 14: Please indicate when and where the consent forms will be signed?

Response: We have added in this text that clarifies obtaining consent. "A consent form will be provided to participants via email, which will be signed and returned before the interview commences. Verbal consent will also be obtained right at the start of interview"

Comment 15: Please clarify how the data will be managed (qualitative and quantitative) to ensure confidentiality.

Response: We have included a short paragraph on data management that reads as below:

Data Management

All data (qualitative and quantitative) collected during the course of the research will be kept strictly confidential and will not be shared with anyone outside the research team. The identity of participants will be known by the researchers who will assign participant identity numbers. Responses will be de-identified prior to analysis, and results will be reported at an aggregate level, so that participants cannot be identified.

Comment 16: Page 7 line 25: What kind of software are you planning to use for the thematic analysis? For example, you could consider NVivo or other. What is your plan for the qualitative data dissemination?

Response: We have now stated the use of NVivo for qualitative data analysis and added a sentence about the data dissemination. It reads "This qualitative data generated will be used to inform the DCE survey to be disseminated to participants in phase II. We will also disseminate the data as presentations in scientific conferences and publications in open access journals."

Comment 17: Page 8 line 27: "Similar to the phase 1 interviews, participants will be purposively selected across different..." see my comment for Page 7 line 15-18 above. And provide sufficient justification.

Response: We have now clarified that we will employ same strategy as with the interviews to ensure diversity in recruited. The revision now reads "Similarly as stated for the interview, we will employ same strategy to recruit disabled people and people from sexual minority groups for the survey to allow sub-analyses of the results". This is to avoid repeating text in the paper.

Comment 18: Page #9 line 50: Seems personal /private, I think you have to re state "Have had sexual health problems, low sexual satisfaction or function in the last five years", maybe you can use the phrase like 'if they have the experience of accessing sexual reproductive health service', prior to this study.

Response: Thank you for the suggestion. This has now been reworded according to read: "Had the experience of accessing sexual reproductive health service', prior to this study"

Comment 19: Page #9 line 56: Change the word "Able" to "willing to"

Response: This has now been revised to read "willing to"

Comment 20: Page #11 line 16: Under “Impact”, please state the impact of the study for the study population or for the knowledge users.

Response: We have now added impact of our study findings on the community and end-users

Comment 21: Page 13: References section require formatting.

Response: This has now been checked and formatting changes applied as relevant.

Reviewer: 2

Christine Curley, University of Connecticut

Comments to the Author:

Comment 1: The need for this study is clear, as sexual health for older adults is often not discussed and understanding the sexual health concerns of older adults is paramount to delivering appropriate, accessible, and effective services. Overall, this study protocol is well thought out and presented.

Response: The authors are thankful for the commendations on the overall project

Comment 2: In the abstract, the two step methods are not clear; it appears that the first stage is the 20-30 qualitative interviews and the second stage is a DCE survey design. Will this be conducted with a second (larger) group of participants or the same 30 participants? Clearly state the components of Phase II in the abstract.

Response: Thank you for point this out to us. We have added a clarifying line at the start of the methods section in the abstract clarifying that the study has two phases before going on to explain the details of the two phases.

Comment 3: Regarding the sample, please indicate how many disabled participants you would be seeking and consider how disability is defined, whether it means physical disabilities, intellectual disabilities or any other disabilities, as the sexual health of these individuals needs may differ.

Response: We have now clarified the estimated size of disabled participants required. Stating this at the end of the Phase II- DCE section. We will aim to have disabled people and sexual minorities make up 10% each out of the total sample size.

Comment 4: In addition, consider whether defining 45+ as “older adults” is appropriate as in most developed countries 45 is closer to middle age. If you have selected 45 as an age where sexual health concerns increase due to menopause, erectile dysfunction or the like, clearly state the basis for this decision.

Response: We have justified the reason for selecting 45 as the cut off age in the last paragraph of the introduction. We also changed the term older adults to ‘middle-aged and older adults’ or ‘adults aged 45 and above’ throughout the paper for better clarity.

Comment 5: On page 4 please “GIVE AN EXAMPLE” as this seems to be missing.

Response: Sorry for the confusion. The example has been provided later in the ‘Developing the DCE design’ section and the appendix file. This sentence has been removed from the background.

Comment 6: Phase II – DCE Survey (p 6) – what is the intended sample size for the described online survey of older adults? Justification for this sample size looks sufficient as discussed on page 9; however it would be clearer if sample size was listed here as well. Pilot sample is discussed on page 8 -is 30 participants; is this sample appropriate and will it be inclusive in the same way as the qualitative Phase I portion?

Response: We will aim to have disabled people and sexual minorities make up 10% each out of the total sample size.

Comment 7: Under “Developing the DCE Design” I am not convinced that 3-5 participants will be sufficient to judge the visual graphics given the fact that the researchers are seeking to design protocols for a range of sexualities, demographics, and ability levels. Perhaps you can be more specific as to who the characteristics of these 3-5 participants.

Response: The authors agree that we would need feedback from a broader range so we have revised the numbers needed to review graphics to up to 10 individuals including disabled people and those from sexual minority groups

Comment 8: On pages 9-10 statistical analysis is clearly described. Please provide similar detail for the qualitative portion of this study. I will note that Table 2 in the supplemental materials appears to be on track.

Response: We have added the methods for qualitative data analysis, stating we will employ a framework approach, transcribing, coding and generating themes as part of the analysis.

Comment 9: This is an important study and I applaud the researchers for undertaking this work.

Response: Thank you for such positive comments

VERSION 2 – REVIEW

REVIEWER	Gebreemeskel, Akalewold University of Ottawa, School of International Development and Global Studies, Faculty of Social Science, University of Ottawa
REVIEW RETURNED	12-Nov-2022

GENERAL COMMENTS	Thank you for revising the manuscript according to the comments. Please see below my further comments for further consideration and improvement of the manuscript. Write “UK” in full in the title. And, please minimize abbreviations in the Abstract. Abstract: Methods and Analysis and Page 4 and 6: Please differentiate the in-depth interview method and the semi-structured interview guide throughout the document. Dissemination: Who are the potential audience, or knowledge users and how it helps them and the target population. Please include a dissemination strategy in the main body of the article. Page #4 Line 8: add punctuation. Page #5 Line 3-6 include citation, and reference. Page #5 Line 50: Objective 2: Indicate the population as “among middle-aged and older adults in the UK” Phase two data analysis phase needs to be precise. Please add more description. In addition to the disability status of the participant, also, it would be help full to indicate how the team will consider the gender mix of the study population. Page #7 Line: Data Management. Page #7 Line 54: Please check out this statement, and whether it is ethically possible. “The identity of participants will be known by the researchers who will assign participant identity numbers.” Page #7 under the data management section, please clarify how ‘quotes’ would be identified in your reports /them analysis or manuscript. Page #9 Adults aAged 45 years or older, restate Page #6 line 10: Your study population is general population ‘adults aged 45 years or older’ living in UK it is not specific to
---

	patients. Please rewrite the subheading “Patient and Public Involvement”. Please indicate how you could facilitate community engagement. What are the criteria for engaging the community organization? Page #6 lines 32-34 include citations for “data saturation” It is unclear how you protect the confidentiality of the data and where you store the hard copies/survey documents and for how long. General comment: Requires professional English proofread After addressing all the comments and when you re-submit, please include the response document as a separate document to track your response easily and accordingly.
--	---

REVIEWER	Curley, Christine University of Connecticut
REVIEW RETURNED	07-Nov-2022

GENERAL COMMENTS	I appreciate the revisions and attention to comments. A few items to edit. In methods: "people with disabilities" not disabled people. Please make this change throughout the manuscript. I am not clear why the authors changed the phrasing with this revision, as language should be person focused not disability focused.
--

VERSION 2 – AUTHOR RESPONSE

Reviewer: 1

Dr. Akalewold Gebremeskel, University of Ottawa

Comment 2: Thank you for revising the manuscript according to the comments.

Please see below my further comments for further consideration and improvement of the manuscript. Write “UK” in full in the title. And, please minimize abbreviations in the Abstract.

Response 2: Thank you for the suggestion. We have replaced UK with United Kingdom in the title now and reduced the abbreviations in the abstract as much as possible.

Comment 3: Abstract: Methods and Analysis and Page 4 and 6: Please differentiate the in-depth interview method and the semi-structured interview guide throughout the document.

Response 3: This is one and the same. We conducted only of semi-structured interviews for this work, and we have revised the manuscript accordingly.

Comment 4: Dissemination: Who are the potential audience, or knowledge users and how it helps them and the target population.

Response 4: We have now revised this section and listed potential stakeholder to target for dissemination including health policy makers, clinicians especially in sexual health and clinical commissioning groups, NGOs and others involved in sexual health service provision.

Comment 5: Please include a dissemination strategy in the main body of the article.

Response 5: This has been added at the end of the main body of the manuscript on page 12, just before authors contribution.

Comment 6: Page #4 Line 8: add punctuation.

Response 6: We are unable to download the PDF file from the editorial system and identify this line. We are happy to revise in the proof stage if the editorial group can help us locate the sentence.

Comment 7: Page #5 Line 3-6 include citation, and reference.

Response 7: Again, we are unable to identify this sentence but happy to insert citations if the editorial group can help us locate.

Comment 8: Page #5 Line 50: Objective 2: Indicate the population as “among middle-aged and older adults in the UK”

Response 8: This has been included now to objective 2

Comment 9: Phase two data analysis phase needs to be precise. Please add more description. In addition to the disability status of the participant, also, it would be help full to indicate how the team will consider the gender mix of the study population.

Response 9: We have reworded the section on purposive sampling to consider varying gender mix of participants being recruited.

Comment 10: Page #7 Line: Data Management.

Page #7 Line 54: Please check out this statement, and whether it is ethically possible. “The identity of participants will be known by the researchers who will assign participant identity numbers.”

Response 10: We will collect some basic demographic details of participants to help our analysis, this demographic data can very well identify participant however we will assign numbers to anonymize the results. We have clarified this in the Data Management section.

Comment 11: Page #7 under the data management section, please clarify how ‘quotes’ would be identified in your reports /them analysis or manuscript.

Response 11: We have clarified and reworded this section as follows: Responses will be de-identified prior to analysis, and results will be reported at an aggregate level, individual quotes when used will be followed by participant assigned numbers so that participants cannot be identified.

Comment 12: Page #9 Adults Aged 45 years or older, restate

Response 12: This has now been corrected to read as suggested

Comment 13: Page #6 line 10: Your study population is general population ‘adults aged 45 years or older’ living in UK it is not specific to patients. Please rewrite the subheading “Patient and Public Involvement”. Please indicate how you could facilitate community engagement. What are the criteria for engaging the community organization?

Response 13: The subheading “Patient and Public Involvement” is as required from the journal and within this subheading, we have clarified criteria for contacting community-based organizations.

Comment 14: Page #6 lines 32-34 include citations for “data saturation”

It is unclear how you protect the confidentiality of the data and where you store the hard copies/survey documents and for how long.

Response 14: We have now clarified that all data will be stored online, this includes interview transcripts and survey responses. We will not be generating any hard copies of data for this project.

General comment:

Requires professional English proofread

Response: Read by a native English speaker.

Reviewer: 2

Christine Curley, University of Connecticut

Comment 1: I appreciate the revisions and attention to comments. A few items to edit.

In methods: "people with disabilities" not disabled people. Please make this change throughout the manuscript. I am not clear why the authors changed the phrasing with this revision, as language should be person focused not disability focused.

Response 1: Thank you for spotting this, we have made this change following consultations and recommendations from the disability research community both on our steering committee and external. The preference is 'disabled people' - see social model of disability here.

VERSION 3 – REVIEW

REVIEWER	Gebremeskel, Akalewold University of Ottawa, School of International Development and Global Studies, Faculty of Social Science, University of Ottawa
REVIEW RETURNED	17-Jan-2023

GENERAL COMMENTS	3RD Version Thank you for revising the document. I have some minor editorial comments to be addressed.  1. Be consistent while using acronyms, the first time you use an acronym, it's important to spell out the full term, Then, you can consistently use the acronym, check the following and edit them in the entire doc: UK or the full term DCEs or Discrete choice experiments 2. Please be consistent on the type of font and size throughout the document. 3. Your response is not what I was expecting. Page 6 line 56-57: The statement is not clear yet, I'm not sure how online data
---

	storage could be secure and confidential. "All data (qualitative and quantitative) collected during the course of the research will be stored online." I would suggest you to rephrase or delete the statement.
--	---

VERSION 3 – AUTHOR RESPONSE

1. Be consistent while using acronyms, the first time you use an acronym, it's important to spell out the full term, Then, you can consistently use the acronym, check the following and edit them in the entire doc: UK or the full term, DCEs or Discrete choice experiments

Response: Thanks. These have been corrected now.

2. Please be consistent on the type of font and size through out the document.

Response: Times New Roman, font 12 are now used for the entire manuscript.

3. Page 6 line 56-57: The statement is not clear yet, I'm not sure how online data storage could be secure and confidential. "All data (qualitative and quantitative) collected during the course of the research will be stored online." I would suggest you to rephrase or delete the statement.

Response: All data (qualitative and quantitative) collected during the course of the research will be stored and managed securely online using our institute's Open Data Kit (ODK) at the London School of Hygiene and Tropical Medicine (LSHTM). An ODK server hosted at LSHTM allows researchers to submit de-identified data electronically from anywhere in the world to a safe and secure server environment that is hosted on our institutional IT infrastructure. This has been added to the manuscript for more clarity.

More information about ODK services at LSHTM can be found here:
<https://opendatakit.lshtm.ac.uk/lshtm-odk-servers/>